**Vulnerability, resilience and adaptation of societies during major extreme storms during the Little Ice Age**

Emmanuelle Athimon [1,2] ; Mohamed Maanan[1*]

[1] Université de Nantes, LETG-Nantes (UMR 6554), BP 81227 - 44312 Nantes Cedex 3-France.

[2] Université de Nantes, CRHIA (EA 1163), - 44312 Nantes Cedex 3-France.

* Correspondance to: Mohamed Maanan (mohamed.maanan@univ-nantes.fr)

**Highlights**

We provide the first historic storms database of the European Atlantic coast.

We assess the impact of the major storms on human vulnerability and adaptation.

We propose a relationship between past climate change and extreme events.

The research proposes new strategies based on the experiences of the past societies.

**Abstract.** This manuscript reviews more than 19691 French historical documents from 14 French archive centers. The results show that 101 extremes storms were recorded including 38 coastal floods. Coastal hazards have forced societies to adapt and develop some specific skills, lifestyles and coping strategies. By analysing the responses of past societies to these hazards, useful ideas may be (re)discovered by today's communities in order to enhance the ability to adapt and develop resilience. Similarly, a thorough knowledge of past climate hazards may allow our societies to recreate a link with territory, particularly through the (re)construction of an effective memory of these phenomena.

**Keywords:** Coastal hazard; Human vulnerability; Society's reactions; Little Ice Age.



## 1 Introduction

Climate change alters the occurrence and severity of extreme events. The frequency of these extremes events and their impacts on humans calls for better integrated risk management. Recent IPCC reports (IPCC, 2014) have shown that climate change would lead to increased risks and

intensity of natural hazards, particularly. Over the past fifteen years, at least three extreme storms causing marine flooding, major impacts and deaths: Lothar and Martin (1999), Klaus (2009) and Xynthia (2010) occurred on the French Atlantic coast. These events have exposed the necessity to create an efficient historical reconstruction and analysis of past storms over a long period in France (Garnier E., Surville F., 2011; Sarrazin, 2012; Athimon et al, 2016).

In the XIV$^{th}$-XVIII$^{th}$ centuries, a storm is mentioned in historical documents if its impacts are major, meaning if they disrupt the manorial levy, induce extraordinary expenses, compromise the salt or agricultural productions, put a population in a dramatic situation that could allow them a tax exemption for example. Although very precise and detailed damage descriptions are rare for the period, texts usually focus on consequences of extreme climatic hazards on the environment,

societies and their activities (Barriendos and Martin-Vide, 1998; Athimon et al, 2016). This studies the human, material, agricultural, environmental and economic impacts of storms and apprehending the vulnerability of past societies. In fact, while studying the socioeconomic parameters of impacts in a quantitative way, it is possible to measure a societies vulnerability (Bradzil et al, 2005 ; Pfister, Bradzil, 2006 ; Bradzil et al, 2010), albeit for the early Little Ice Age

(XIV$^{th}$-XVI$^{th}$ c.), this is difficult to achieve. Furthermore, through cultural history, many researchers in historical climatology have recently focused on climate and extreme climate hazard representations, including people's attitudes and reactions (Pfister, 2010; Gerrard and Petley, 2013). As defining vulnerability of late medieval and modern societies is not the point of this paper, we will not discuss it. We will only admit the concept of vulnerability induced indicates

both the impacts of climate hazards on populations and the societies' ability to anticipate, mitigate

and adapt (Fussel, Klein, 2006).

Though the Little Ice Age (LIA) was a cooler period (Le Roy Ladurie, 1967; Lamb, 1991; Pfister, 1996), this research aims to show how past societies can constitute a source of inspiration for present communities confronted with the effects of global warming, increasing vulnerability and

risks. A wide body of research has proved that during the LIA, there was an increased frequency in the number of storms (Brooks, 1949; Gottschalk, 1971-1977; Lamb, 1980 and 1985; Hickey, 1997; Galloway, 2009) and, more recently, based on the number of proxies, Trouet et al (2009) provide new evidence for enhanced storminess in NW Europe during the LIA. Similarly, Vide J.M. and Cantos J.O. (2001) in *Clima y Tiempos de España* note four periods of catastrophic events (mid-

15th century, 1570-1610, 1769-1800 and 1820-1860) marked by heavy rains, snowfalls and sea storms. Moreover, in Western Europe, extreme storms can be devastating. In the North Sea, the sea floods of the Danish, English, Dutch and German coasts in 1421 and 1446 killed more than 100,000 each in England and the Netherlands, while those of 1570 caused over 400,000 deaths (Lamb, 1991, p. 174).

This paper is aimed at helping communities adapt to climate change by outlining a set of lessons derived from coastal hazard experience together with insights from contemporary and modern scholarship. These lessons are framed in the context of building resilient and sustainable communities in this era of climate change. This step will be followed by the development of a database on extreme events over the past thousand years, to assess the impact of past climate

changes on changes in the frequencies and intensities of these events.

From historical examples of storms during the Little Ice Age, this research will:

1) Identify, as accurately as possible, the storms and the coastal floods on the French Atlantic coast,

2) Discuss the impacts of these phenomena on populations,

3) Define the vulnerability, resilience and adaptation of French medieval and modern societies by

presenting their perceptions and their reactions.

The geographic framework of this study is located in France, from Brittany to Gascony (figure 1),

while the period of study stretches from the XIV$^{th}$ into the XVIII$^{th}$ century. The study area includes

rocky and sandy coasts, lowlands and marshes, and islands. This area is regularly affected by

violent winds and episodic coastal flooding (Feuillet et al., 2012; Sarrazin, 2012; Garnier et al.,

2018; Pouzet et al., 2018a; 2018b).

## 2 Materials and Methods

This research is based on French historical published and unpublished documents including

narrative sources (chronicles, diaries and memoirs), archives (records of city repairs, books of

accounts, parish registers, and surveys conducted after a disaster, etc.) and ancient maps. In all,

19691 documents were consulted within the archive centers of the 14 French cities of: Nantes,

Angers, Saumur, Le Mans, Laval, Rennes, Vannes, Brest, Quimper, Saint-Brieuc, La Roche-sur-

Yon, La Rochelle, Bordeaux and Paris. These historical documents contain observational and

descriptive data on past storms, their impacts on societies and the reactions and adaptation to them,

allowing the study of past societies' vulnerability. Every historical source used for the redaction of

this paper, including those already studied and published by other scientists, have been consulted.

They have all been analysed and criticised.

The critical approach of sources is crucial. The redaction conditions of a document are essential to

consider, as is the author and the institutional framework. No descriptive meteorological data are

objective. Depending on the writer's age, his social environment, his life experiences, memories,

perception of phenomenon, his propensity to exaggerate etc, the event's description will change. It

is also necessary to define if the document is contemporary and if the author witnessed the events related (Hickey, 1997). If not, it needs to be determining whether the events occurred during the life-time of the author or if he copied it from other documents. In the case of a copy, the original

manuscript (if it still exists) needs to be consulted to make comparisons and eventually correct any errors, mostly of date, in the transcription. Particular attention should be paid to dates (Bràzdil et al., 2005). In the West of France during the XIV$^{th}$-XVI$^{th}$ centuries, Easter style dates predominated. Therefore, the New Year started at Easter. Here, dates have been homogenized in order to appear in the « new style (n.st.) ». In other words, the « new » year is meant, even when the event happens

in February. Moreover, in 1582, Pope Gregory XIII created the Gregorian calendar. This replaced the Julian calendar, which was then counting 10 days less (of late on the sun). In this present paper, before 1582, dates will be given both in the Julian and Gregorian calendars, the Gregorian dates being in parentheses. The correction of Julian dates is possible by adding 8 days to the XIV$^{th}$ century dates, 9 days to those of the XV$^{th}$ century and 10 to the XVI$^{th}$ century's dates. The

possibility of improper understanding and interpretation of sources and data by the researcher should finally be admitted (Bràzdil et al., 2005).

According to Le Roy Ladurie (1967), to reduce the risk of improper interpretation and to use data contained in historical records for the study of climatic events and effects, four conditions must be satisfied: 1) the series must be annual; 2) they must be continuous, so without documentary gaps,

3) homogenous and 4) quantifiable. In matters of historical reconstruction of storms in the Little Ice Age, this methodology cannot exactly be followed. However, the important difference of the quality of documentary sources necessitates making a selection of data. This research is based on a strict critical approach of sources as discussed earlier and on precise criteria to generate data that is as reliable, coherent and relevant as possible (Barriendos and Martin-Vide, 1998). For this

research, the identification and authentication of a storm relies on four criteria:

1) The information comes from an eyewitness or a contemporary of the event[1]. However, the data are taken into account only if other first-hand sources and/or bibliographical documents confirm the reliability of the information.

2) The event is presented in the historical records as a storm and not, for example, as strong winds or a thunderstorm with high winds.

3) The date, as accurate as possible, is recorded, either because it is specified on the original document or it is possible to estimate it.

4) There is a description of the damage caused.

Once these four criteria are fulfilled, information is scrupulously analysed, criticized and included in databases to produce cross-checking. The cross-checking and comparison of data from different documentary sources allows the historical reconstruction of storms within a defined temporal and spatial frame.

Fundamentally, the process of reconstruction is nevertheless difficult: some limits to the storm reconstruction work must be taken into account and discussed. First, as explained in the results section below, storms recorded in historical documents are ones with major impacts, even if climatologically speaking these are not necessarily the most violent events. It is therefore obvious that for many of these hazards, the historical climatologist has no information and even no clue they happened. In these circumstances, the reconstruction is somewhat biased. Secondly, the series of storms recorded in the West of France are discontinuous owing both to an oral tradition (Sarrazin, 2012) and many documentary gaps (Athimon, 2017). They are due to historical

---

1    The original documents were produced by intellectuals, clergymen, people with municipal responsibilities, etc., with a good level of education and culture. So, the information tends to be reliable, though sometimes exaggerated.

contingencies such as archival disasters[2], the French Revolution[3], wars such as the Hundred Year War against England in the XIV[th]-XV[th] centuries, the religious wars between Protestants and Catholics in the XVI[th] century; the Second World War etc. The loss of these documents certainly deprives the searcher of precious censuses of wind hazards. These first two limitations make any

reliable appreciation of the recurrence of storms difficult. Moreover, today's satellite images allow precise delineation of the wind direction, the trajectory of a storm and the flooded areas, during medieval times a precise spatial demarcation is impossible to establish, mainly due to a lack of information, documentations and instruments. Finally, the descriptions are often either exaggerated or undervalued, which makes the characterization and the intensity estimations of a phenomenon

difficult to undertake for the XIV[th]-XVIII[th] centuries.

Despite these limitations, through an acute critical study of documents, selection criteria of data and cross-checking, the historical climatologist can identify and characterize noteworthy storms and sea floods of the past.

**3. Results and discussion**

**3.1 Reconstruction of past storms**

With 19691 documents studied from XIV[th] to XVIII[th] c., this research is the most exhaustive work done on storms and storms causing coastal flooding during the Little Ice Age on the French Atlantic coast.  138 storms were identified from the beginning of the XIV[th] century to the end of the XVIII[th] century. However, due to the application of the strict criteria presented above, the final list

of these phenomena is not as high in number and only 101 occurrences could be selected.

Figure 2 establishes a trend of increased and decreases storm activity. As can be seen, for instance,

---

2        For instance, the Chamber of Accounts fire in Paris in 1737 and the fire at the Municipals Archives in Bordeaux in 1862 have caused the destruction of entire sets of registers, bundles and acts.
3        Almost all of the documents relating to the life, history and possessions of some noble families, such as the Rohan and the La Trémoïlle families in Brittany, disappeared during the revolutionary episodes.

the late XVI[th] century shows an intense stormy period. This period between 1560/70-1630/40, is well-known by historical climatologists and has been named the "Second Hyper LIA" (Lamb, 1980; Le Roy Ladurie E., 2004). Similarly, the first twenty years and the end of the XVIII[th] century

seem to stand out (Lamb, 1991). (However, it may be linked to a larger number of historical documents.) I don't understand what you want to say here? In fact, owing to the absence of documents this figure is not exactly representative of the reality of the storms studied in the space-time frame, especially for the earlier periods. Furthermore, in Aunis, particularly Ile de Ré, the area on the border of Brittany and Poitou, called "Marches" (Bouin, Bourgneuf and Noirmoutier

island) are well represented (figure 3). This can be explained both by extensive research on this spatial framework (Sarrazin, 2012; Athimon et al, 2016; Garnier et al, 2018) that a natural vulnerability exists?–these coastal areas are low lying, sandy and/or marshy (Pouzet et al, 2018a; 2018b). Conversely, Brittany, which is subject to violent winds and located on the West-East wind trajectory, seems not to have been badly affected by past storms. This is mainly explained by the

loss of a large number of archives during the wars and French Revolution mentioned previously.

Once they are identified, analysis of the storm descriptions allows the study and definition of the impacts of storms on societies and their activities.

**3.2 Impacts, damage caused by storms**

Most of the damage was caused to monuments and buildings (figure 4). Damage to infrastructures

is globally demonstrated in a precise hierarchical structure. First are the religious monuments, whose alteration by wind (disrupts populations). How? Second, the constructions where repair costs affect local finances are mentioned. The final cluster shows, noble houses while modest homes are often ignored. For instance, a violent storm hit the French Atlantic coast on June 24[th] (3[rd] July) 1452. It knocked down the bell towers of two churches in Angers (De Bourdigné,

*Chronique* written in 1529) and washed away part of the roof of the La Tremoille family's castle





on Noirmoutier island (AN[4] Paris, 1 AP 1976, n°176).

Storms can also lead to major damage on agriculture: damaged crops, seed dispersal etc. On July 2nd (12th) 1507 a storm occurred. One of the effects of the violent winds was the destruction of many wheat fields in Ancenis, Nantes and the surrounding areas (AD Loire-Atlantique, E 269,

second book, f°8). In cases of significant agricultural damage the manorial levy could be disrupted. For instance, in 1346, the sacristan of Bourg conducted an investigation in the parishes of Bayon and Gauriac in Gascogny. Farmers had difficulty paying taxes due to storm damage (AD Gironde, GG 236-II). Agriculture damage aside, a violent wind can also spoil the fauna and the flora: stranded fish, damaged trees and so on. From December the 29th 1705 to January the 1st

1706, there was a severe storm on the north-west French Atlantic coast. Brittany, and the countryside of Mayenne, Sarthe and Normandy, suffered extensively. During this storm, just on the property owned by Saint-Melaine Abbey in Rennes Do you mean "just one property survived"?, more than 120 trees were uprooted (Bordeaux, *Journal d'un bourgeois de Rennes*, 1598-1800 , written by 5 eyewitnesses, 1992 edition), while in the parish of Montjean and that of Courbeveille

for example, more than two thousand fruit trees, chestnut and oak trees were uprooted (AD Mayenne, E dépôt 60/E13, view 6-7 ; E dépôt 116/E9, view 103).

This kind of extreme climate hazard can also endanger people's lives. Sources registering the number of dead as a result of a storm or a storm with sea flooding are rare. Descriptions mostly remain using words such as "several", "many" and "countless". From time to time, a source stands

out and gives a number. It should nevertheless not be considered as the total number of deaths during the event since information is always much localized. According to Etienne De Cruseau, at the end of March 1591 (end of February in reality), a west-south-west storm hit the French Atlantic coast. In Bordeaux structural damage was so significant that at least 4 people died in their homes, while several were injured by falling roof tiles (De Cruseau, *Chronique*, written between

---

4    AN: National Archives; AD: Departmental archives; AM: Municipal archives.



1587 and 1616, 1879-1881). Similarly, on November the 3$^{rd}$ 1656, during a violent storm which

caused widespread damage on the Poitou, Brittany and the "Marches" coast, at least 184 people

died in shipwrecks, such as the Marshal of Meilleraye warship (AM Nantes, GG 485, f° 87).

During the medieval and modern periods, no overall or itemized records on the economic or

human toll were provided after a climate event by the different institutions (urban, seigniorial, and

royal). This can be explained as much by the thinking of the day – the very notion of a human toll

was unknown, as by the authorities' lack of organisation, the poor administrative procedures, and

other means available. A general estimation of the economic and human cost of storms is therefore

unrealizable. For a limited spatial framework such as property, salt marshes, a fifedom, a town

district, etc. accurate and reliable information on the amount of human (cf. supra), material or

economic losses or on the cost of repairs can nevertheless be made available. Thus, in 1469 or

1470[5], owing to flooding, the Blanchet family, modest notables, lost eight (heaps) Is this a

measurement of salt? of salt in the salt marshes of Bourgneuf and Prigny (AD Loire-Atlantique, 2

E 382), which is equivalent to the annual income of an average seigniory (Athimon et al, 2016).

Sometimes information is not quantified, but the climate hazard is so extreme that several years of

cleaning, restoration and repair are required. This kind of data thus gives interesting estimations of

the loss generated by the event. During the winter of 1351-1352(n.st.), a storm with coastal

flooding overwhelmed some areas, in particular Noirmoutier island, for nearly half a century (AN

Pierrefitte-sur-Seine, 1 AP 1974, n°50) and destroyed salt marshes in Olonne (*Cartulaire de

l'abbaye de Saint-Jean d'Orbestier*, 1877). Damage was so severe that almost 15 years later, in the

middle of the 1360's, salt production had still not been re-established.

Where possible – depending on the data stored in historical documents –, the study of past storms

and their impacts must be followed by an analysis of societies' reactions and responses to these

---

5        The destruction of these salt heaps is either due to the storm with sea floods on 27-28$^{th}$ of January (5$^{th}$-6$^{th}$
February) 1469(n.st) or an unknown event which occurred between the end of the spring and the beginning of the
summer of1470.



kinds of extreme climate hazards. The question of past societies' vulnerability, their adaptation and resilience is of huge importance. This appears to be the new focus of interest for historical

climatologists (Pfister, 2010).

**3.3 Societies' reactions and responses**

Constantly living with the risk of storms and sea floods, coastal societies have developed significant risk awareness, an effective memory of these extreme climate hazards, a specific way of life, a particular perception of natural hazards and risks. In fact, storms and sea floods were part

of the culture and habits of past coastal societies, who were used to dealing with them and considered them a normal part of life. This is precisely what enabled them to fix memory and develop some specific knowledge. The most destructive, traumatic or atypical storms and sea floods were preserved in their collective memory. The preservation and dissemination of these memories over a long period developed risk awareness among ancient societies. In 1627,

inhabitants in Bouin sent a petition to king Louis XIII in which they certified having suffered from more than 15 storms and coastal floods since the year 1500 (Luneau et Gallet, 1874, n°XXXIII). Furthermore, they wrote another memorandum dated 18[th] November 1775 in order to defend and explain their privileges in which they mention the storms and sea flooding which affected Bouin on November the 13[th] (23[th]) 1509, December the 31[th] 1598, September the 7[th]-8[th] 1599, March the

14[th]-15[th] 1751 (AM Nantes, II 136, n°30). This document is however not completely reliable. There are several limitations: the memory is sometimes fragmentary and selective. This can result in exaggerations, anachronisms, mental reconstructions etc. It also relies on the author: his age, his discernment, understanding and interpretation of the events, etc. (Athimon et al, 2016). A climate event can also seem more violent than an earlier identical one. Despite these limitations, memory

of extreme climate hazards can continue over several decades, which results in promoting risk awareness, develops adaptability and resilience and a better understanding of the elements. A better understanding and interpretation of the elements is of significant importance. Unlike today,

ancient coastal societies considered the sea a constant and unpredictable threat. In 1451, a memorandum, sent to the court of King Charles VII, sets forth the feelings of coastal societies and

its dangers (*Mémoires présentés au roi Charles VII par les délégués de la ville de Poitiers pour le détourner d'établir la gabelle en Poitou et en Saintonge*, 1873). This document rests on an ancient, but reliable, hazards observation practice and a perpetual struggle with limited resources against the sea and its risks. In the XIV[th]-XVIII[th] centuries, risks and storm hazards were incorporated into memory, culture, habits and the lifestyle of coastal societies. This contributed to the development

of specific knowledge, risk awareness and prevention measures.

During the studied period, dikes are built in order to provide sea defences against swell, chop and tidal movements, and (gain lands on the sea) Do you mean "to reclaim land from the sea"? (Thoen et al., 2013). They also protect salt marshes, fields, infrastructures and local populations from coastal flooding. They are small, around two meters high, and made up of clods of clay, sand,

wood and pebble (Sarrazin, 2014). Leaky and fragile, they are easily destroyed but are useful coastal defences. Moreover, their structure makes them very easy and quick to repair. Unlike the Netherlands (Soens, 2009) or England (Galloway, 2009), on the French Atlantic coast no real authority interferes in dike construction (Sarrazin, 2014) – (although the royal authority has gradually tried to intermediate (Boucard, 2010)) I cannot verify this reference. Dyke construction

and maintenance relies either on owners of lands and salt marshes or, if they are public, on the community (Sarrazin, 2014). In 1492-1493(n.st.), some dykes broke under sea and wave pressure on the island of Noirmoutier. Their restoration cost was modest for lord La Trémoïlle, owner of these dikes (AN Pierrefitte-sur-Seine, 1 AP 1964). These low cost are mostly due to the abundance of cheap labor. However, sometimes the cost is exorbitant. For example, in a document dating

from the 7[th] of December 1663, David Tessier and his wife, Janne Jumel inhabitants of Croisic, near Guérande, state that they had to restore their marsh dykes following a violent storm on January the 12[th]-13[th] 1663. These repairs cost more than 6000 livres (AD L-A., B 655, f°277)!

The vulnerability management of past coastal societies is mainly based on a precise social and work organization. Everyone (lords, farmers, fishermen, salt producers etc.) has a specific role to play. Everyone's involvement in the construction/repair of dikes and territory management develops risk awareness and ensures its dissemination within society (Sarrazin, 2014). The risks are then perfectly integrated into the coastal societies' conception and lifestyle (Galloway, Potts, 2007). Therefore, the vulnerability of coastal societies, in particular salt producing ones, is scaled down. Furthermore, during the late Middle Ages and modern period, dikes are built in a subdivision system. In other words, it is an intricate network of levees (figure 5). Next to Anse de l'Aiguillon, a map of Champagné swamps drawn by André Chevreux in 1656 (AD Vendée, 1 E 442) presents this spatial subdivision. Dikes follow each other in a quasi « sequential » order: the newest are erected on the seafront, while the older ones are further back. This technique is part of a preventive approach; (even if it does not come from an upstream reflection) I don't understand what you mean here. (Athimon et al, 2016). When the sea floods, it encounters a multitude of small levees which absorb the wave energy and break the speed of waves, lessen the intensity of flooding and reduce the spread of the water. In fact, populations were aware of the importance in maintaining levees, including the older ones, as noted in a Champagné's seigniorial court document of the 7[th] of November 1560 (Médiathèque de Niort, fonds La Fontenelle de Vaudoré – Clouzot, 1904; Sarrazin, 2014). Finally, the construction, development, maintenance and repair of dikes are part of the lifestyle and the culture of coastal societies. They are also connected to a notion of anticipation of future damage and risk prevention (Athimon et al, 2016).

It was not until the late XVIII[th] century that the royal authority becomes a systematic reference in the case of an extreme climate hazard (Favier, 2002). However, at the end of the Middle Ages and during the whole modern period, French kingdom authorities' interference gradually increases. A range of societies' reactions and responses to climate disaster are seen. Figure 6 shows that the first to react to a disaster is the local population followed by the local authorities, either seigniorial or



urban. Next, the counts, dukes and princes can offer their assistance. The king appears is the last resort. Each « player » offers different reactions which result in three main effects: adaptation,

resilience and a reduction in vulnerability (figure 6). In the case of local authority intervention, actions were mainly practical: materials labor, funding and supervision in order to rebuil and repair. Through court proceedings, local authorities could also compel people to damaged sites. This is what happened after the violent storm with sea flooding on August 22$^{nd}$ (2$^{nd}$ September) 1537 at Ile de Ré : people who did not take part in the cleaning, restoration and repair of flooded

areas or public levees could incur very heavy fines (AN Pierrefitte-sur-Seine, 1 AP 2002). In addition, ducal and royal institutions could also interfere. Their reactions are mostly financial: they provide tax assistance. In 1392, King Charles VI gives a tax exempts to the inhabitants of Noirmoutier island who fought to prevent the English from landing on the island and regularly experience storms and sea floods (*Recueil des documents concernant le Poitou contenus dans les*

*registres de la Chancellerie de France*, 1893). Moreover, as stated above, the royal authority gradually intermediates on the construction, maintenance and repair of dikes. For instance, on February 9$^{th}$ 1510 (n.st.), after the stormy winter of 1509-1510 (n.st.), and the sea flooding of November the 13$^{th}$ (23$^{th}$) 1509, the duke of Brittany, also King of France, Louis XII sends a letter to his alderman in Nantes to coerce him to visit the disaster areas (AD Loire-Atlantique, B 20, view

34 numerisation). The aim is to keep abreast of the situation, have an idea of the amount of damage and see if repairs are needed. The damage is so significant that on the 23$^{rd}$ of June 1511, the duchess of Brittany, also Queen of France, Anne, orders a 5 year tax exemption to the disaster victims (Luneau S., Gallet E., 1874). Extreme climate hazards such as storms and sea floods have led the various authorities in the kingdom of France to take decisive action. Their point was

somehow to provide post-disaster support and prevent future risks.

**4 Conclusion**

Historical documents contain data on weather disasters (and climate in general), so research in

historical climatology can identify past storms over time, the impact of these phenomena on populations and responses, adaptability and resilience of ancient societies. By providing

information about past extreme climate hazards, this field is of the utmost importance for the current concerns and discussions about climate change, increasing vulnerability and risks, and greater severity of climate hazards. Indeed, a thorough knowledge of past storms and coastal flooding allows today's societies to: 1) recreate a link with their territory, 2) develop an effective memory of these events, and 3) propose new strategies based on the experiences of ancient

societies. Historical research can make a relevant contribution to prevention and management of natural risks and hazards.

Future works will take into account a greater quantity and a wider range of sources in order to improve the knowledge and reconstruction of past storms. They will be extended over a longer historical period in order to try to calculate the frequency of storms and coastal floods. Another

objective is to identify the periods of increased and decreased storm activity over the last seven hundred years and correlate them with climatic fluctuations (Lamb, 1980). This may also enable us to identify modifications in attitudes and lifestyles, some changes in risk perception, loss of habits, technical development and their connections to adaptability, resilience and vulnerability. By analysing the responses of past societies to extreme weather hazards, useful strategies may be

(re)discovered in order to enhance the ability to innovate, adapt and develop resilience in today's societies. In collaboration with French researchers, some correlations will also be made with coastal sediment. Finally, with the aim of identifying trajectories, storm corridors etc., international collaboration will be required ; including co-operation with sociologists, psychologists, anthropologists, etc., which will be of great interest to investigate and better understand the social

perceptions and representations of past extreme climate hazards (Brazdil et al, 2005).

Open Access EGU

**Acknowledgments**

Analysis, operating costs and field measurements were funded by the Fondation de France (contract number 1503) and OR2C-AXIS 3 (part of the OSUNA program and the Pays-de-la-Loire region).

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

**Primary sources**

• Handwritten documents

Municipal Archives of Nantes :

GG 485, f°87

II 136, n°30

Departemental Archives of Gironde :

G 236-II

Departemental Archives of Loire-Atlantique :

B 20, view 34 (numerisation)



B 655, f°277

E 269, second book, f°8

2 E 382, f°4

Departemental Archives of Mayenne :

 E dépôt 60/E13, view 6-7 (numerisation)

E dépôt 116/E9, view 103 (numerisation)

Departemental Archives of Vendée :

1 E 442

National Archives of Pierrefitte-sur-Seine :

1 AP 1964

1 AP 1974 - MIC/1AP/1974, n°50 (microfilm)

1 AP 1976 - MIC/1AP/1976, n°174 (microfilm)

1 AP 2002 – MIC/1AP/2002, n°75 (microfilm)

Municipal Library of Angers :

Oudin G., *Extrait d'un manuscrit de Messire Guillaume Oudin, prestre sacristin de l'abaye de Nostre Dame du Ronceray, depuis l'année 1447 jusqu'en l'an 1499*, copy of the XVI[th] century, original lost, ms. 0976 (0858) (numerisation).

• Published sources



Bordeaux Cl., *Moi, Claude Bordeaux : journal d'un bourgeois de Rennes au 17$^e$ siècle,* published by Isbled B., éd. Apogée, Rennes, 1992.

*Cartulaire de l'abbaye d'Orbestier,* (1877), *Archives Historiques du Poitou,* éd. L. de La Boutetiere, t. VI, Poitiers, n°194.

Clouzot E., (1904), *Les marais de la Sèvre niortaise et du Lay du X$^e$ à la fin du XVI$^e$ siècle*, Paris-Niort, pièce n°XVII, p. 237-240 - Médiathèque de Niort, fonds de La Fontenelle de Vaudoré, carton 144, n°4.

De Bourdigné J., (1529), *Hystoire agregative des Annalles et cronicques Daniou,* éd. Galliot du Pré, Paris.

De Cruseau E., (1879-1881), *Chronique*, published by Société des bibliophiles de Guyenne, t. 1 and 2, Bordeaux.

Luneau S., Gallet E., (1874), *Documents sur l'île de Bouin*, Nantes, n°XIII and n°XXXIII.

*Mémoires présentés au roi Charles VII par les délégués de la ville de Poitiers pour le détourner d'établir la gabelle en Poitou et en Saintonge*, (1873), in *Archives Historiques du Poitou*, tome II ,

Poitiers.

*Recueil des documents concernant le Poitou contenus dans les registres de la Chancellerie de France,* published by Guérin P., impr. Oudin, tome VI, Poitiers, n°DCCLXXIII, p. 88-92.



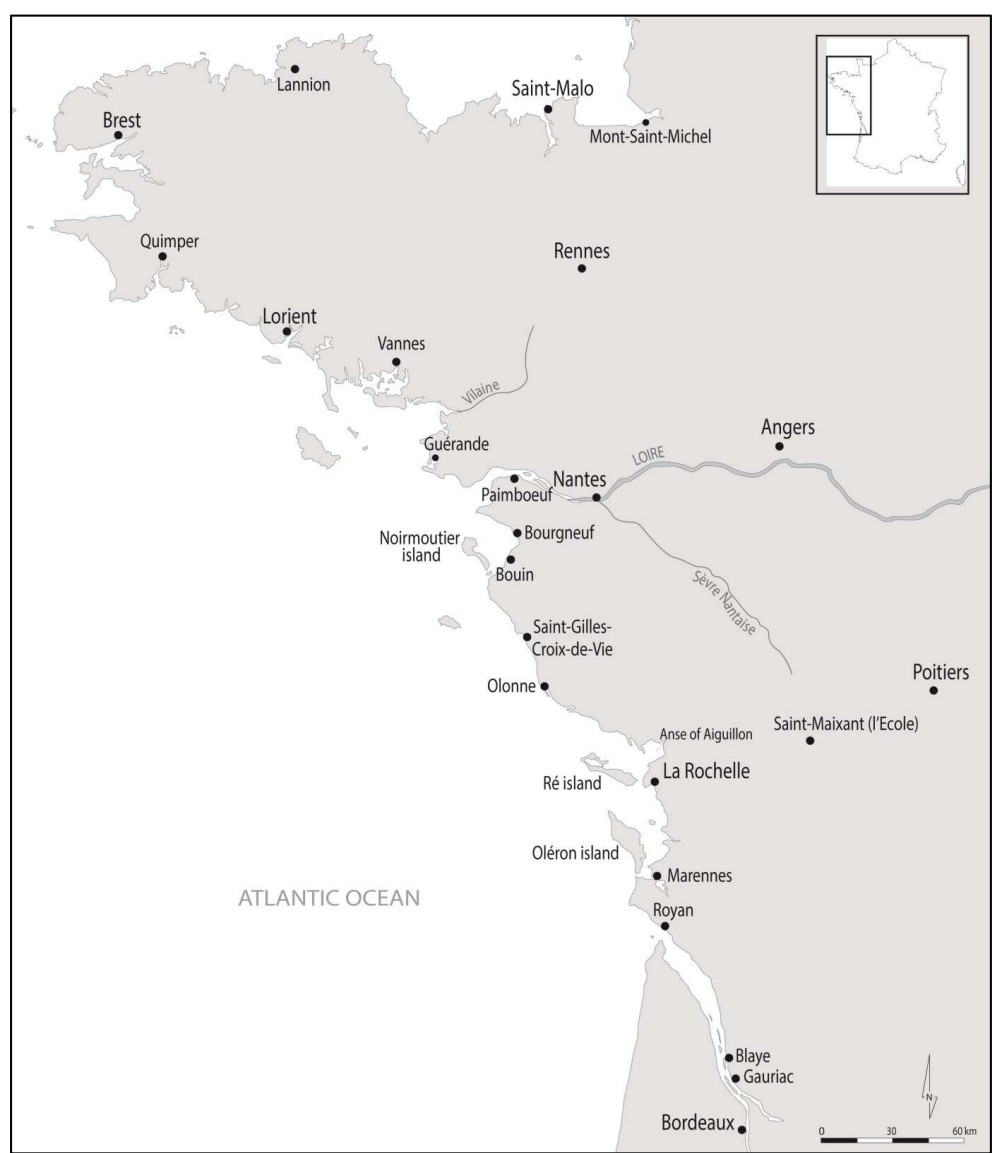

490                                     Figure 1. Location Map

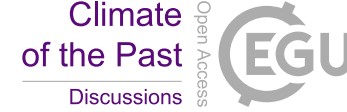



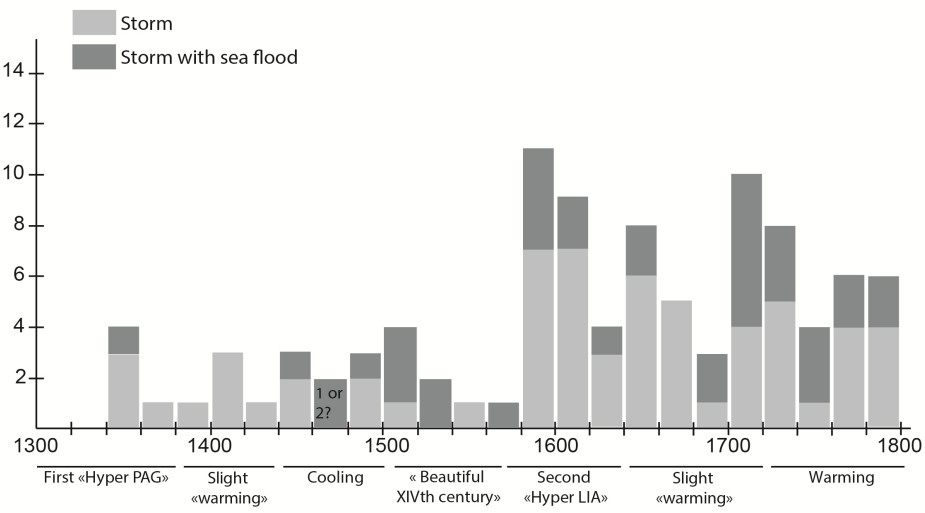

Fig. 2: Storms and storms with sea flood in the study area





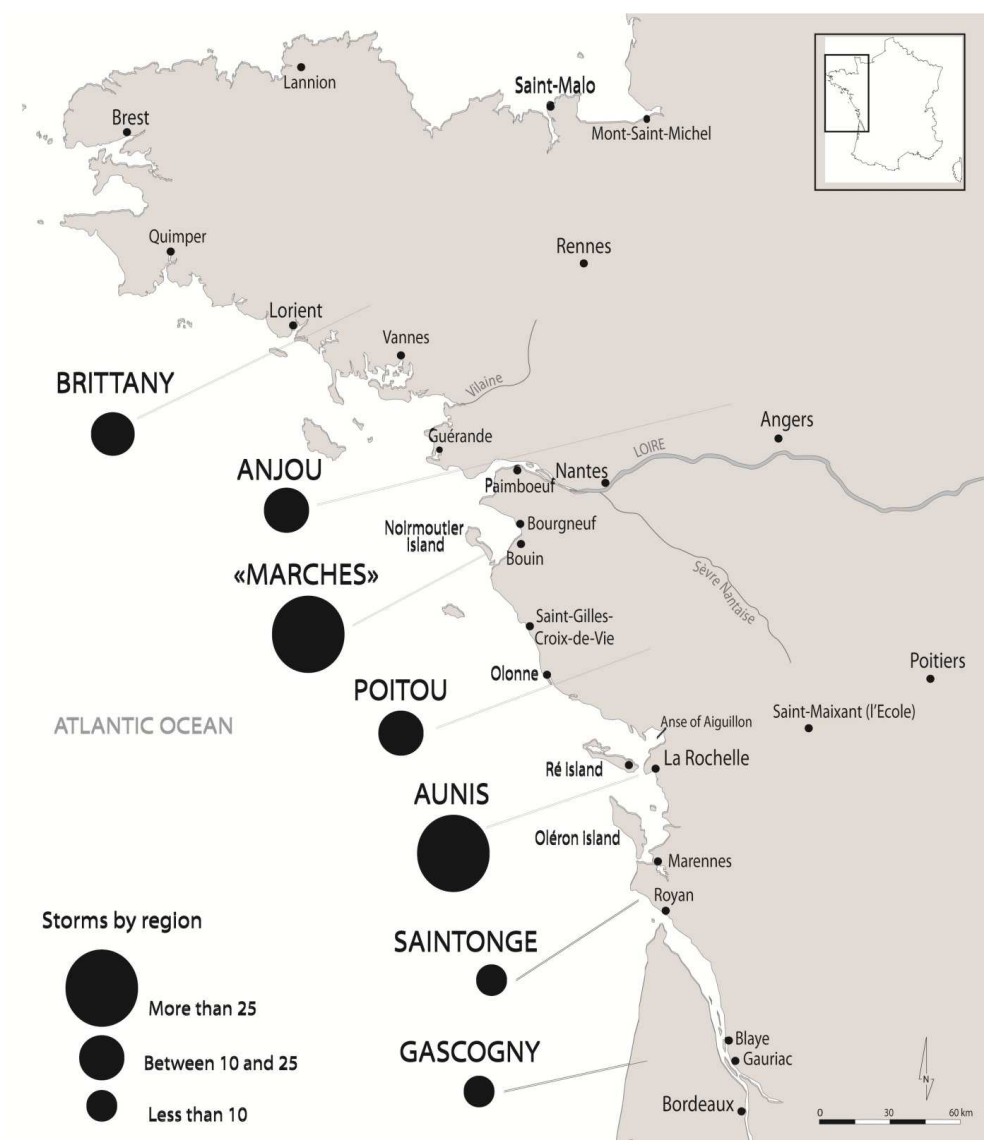


Fig. 3: Map of areas subject to the risk of storms from 1300 to 1800




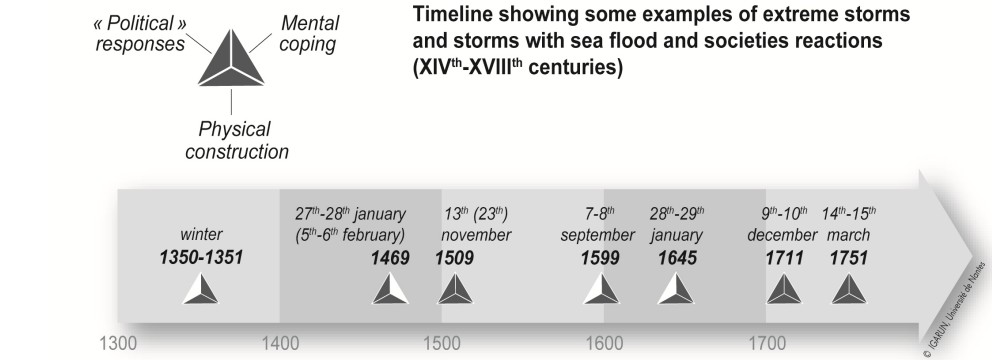

Fig. 4: Major storms and the societies' reactions





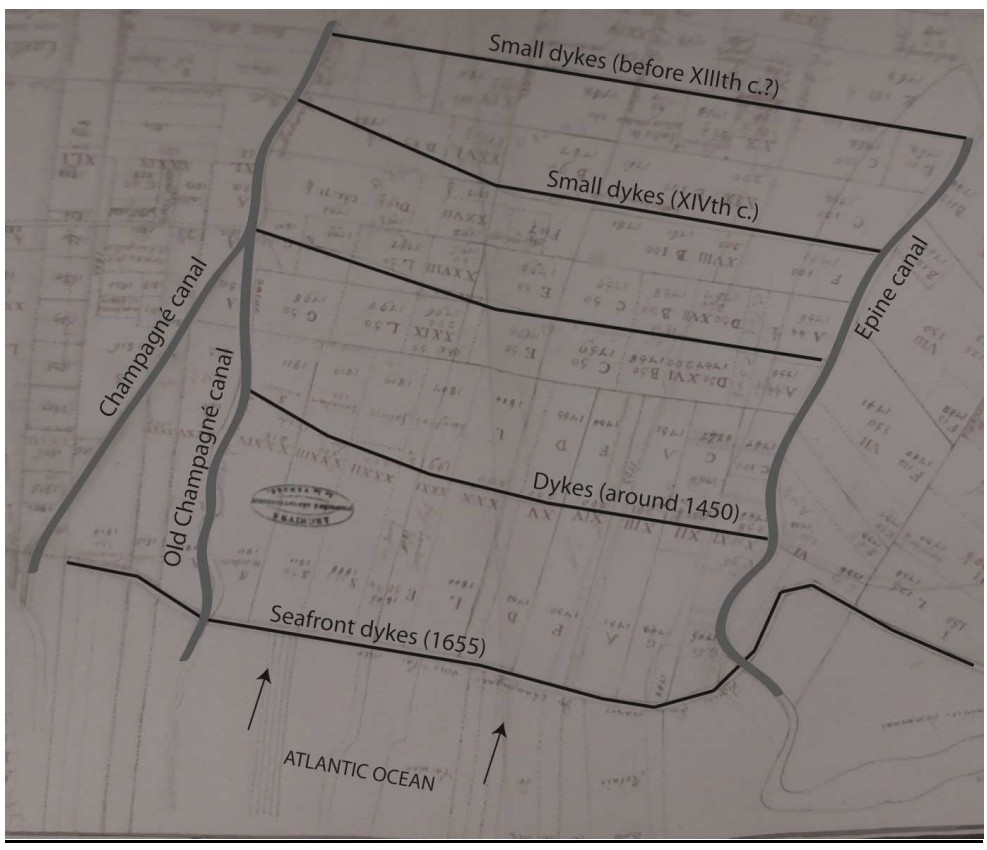

505        Fig. 5: Map of Champagné swamps, André Chevreux, 1656, AD Vendée, 1 E 442, modified





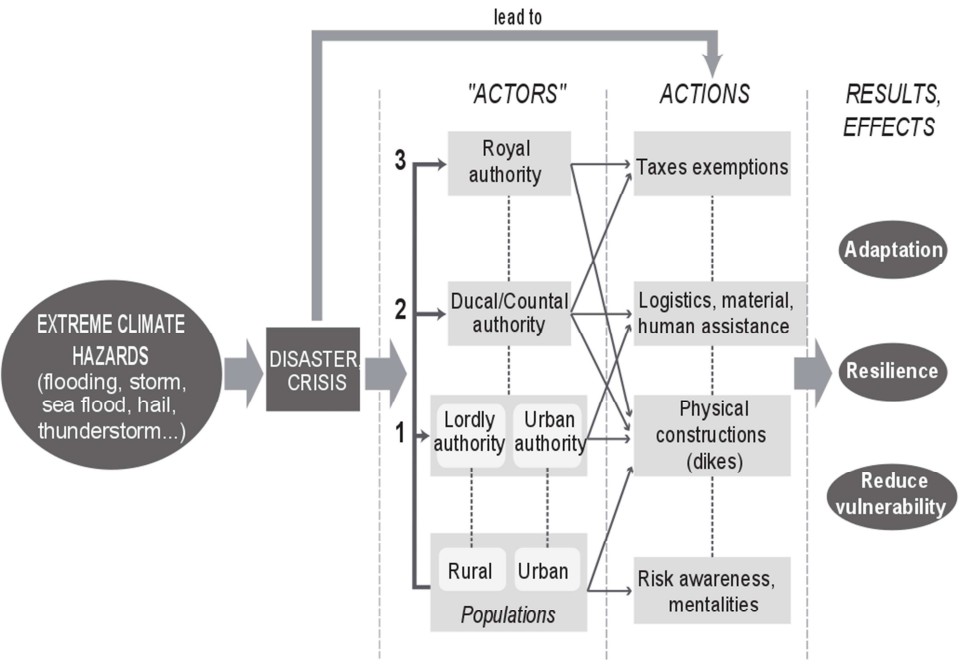

Fig. 6: Simplified representation of governance and disaster management
