# Peer review of "Vulnerability, resilience and adaptation of societies during major extreme storms during the Little Ice Age"

_Climate of the Past, 2018_

## Referee Comment (RC1) · A. El M'rini (Referee) · 20 Jul 2018

The paper "Vulnerability, resilience and adaptation of societies during major extreme storms during the Little Ice Age" deals with a topic of global interest, and aims at the reconstruction of extreme events on the French Atlantic coast during the Little Ice Age, the risk perception of people of this region and their adaptation strategies. In general, I find that the approach adopted by the authors is original, with a huge documentation work (review of 19691 French historical documents from 14 French archive centers according to the authors), which allows the authors to provide the first historic database of storms dating back more than 700 years. So I find this work interesting and contains

all the necessary to be published in the journal "Climate of the Past". However, I have some minor remarks to be considered in order to improve it. - First, Instead of the term ''climate hazard" I prefer the term ''meteorological hazard" when authors talk about storms. - At line 49, authors talk about global warming as a threat that present people have to face, and how the ancient peoples (in this case the people who lived during LIA) could be a source of inspiration for present communities. Then, at the line 50, authors say that the LIA (which is a cold period) was characterized an increased frequency in the number of storms. Can the authors explain how warming and cooling could both lead to an increase in the number of storms?

- The conclusion needs to be further developed; it is elaborated rather like perspectives, and does not specify the main finding of this paper and if the objectives were reached.

- Authors speak about the perception of risk and collective memory, and that the experience of the ancient people could be a source of inspiration for recent societies. However, when they relate ''Societies' reactions and responses" they do not put the link with the recent time, and if the collective memory still active, especially knowing that this region was recently damaged by storms that caused deaths (cf. Xynthia).

Finally, the reconstruction of the past extreme events dating back centuries from documents remains a very delicate task, and I endorse the authors when they concluded that there will be other works using other proxies and correlations with coastal sediments to assert.

---

## Author Comment (AC1) · 11 Aug 2018

Dear Referee El M'rini A.,

First of all, thank you for your review on our paper "Vulnerability, resilience and adaptation of societies during major extreme storms during the Little Ice Age".

You are absolutely right about the term "climate hazard", better use "meteorological hazard". We will change all occurrences of the term.

We will also develop the conclusion and add the main finding of the paper.

Considering the question of current societies and whether the collective memory is

currently active or not, this part of the research is still ongoing. Some colleagues do their PhD on such a subject and we started a few months ago to study it too. However, our results are not as relevant as those of the past period yet, so a pertinent comparison will be part of a futur paper. Perhaps we should expose it more as a near prospect or as operational information for territorial and political leaders.

Finally, with regard to the formation of storms in Europe, it is not likely prooved that global warming will increase the frequency of storms. According to IPCC reports (2014), global warming is likely to increase the frequency of droughts, heavy rains and floods. However, an increase in the frequency of storms is still under discussion. If the IPCC considers this could probably be the case for tropical cyclones (hurricanes, cyclones, typhoons), which appear mainly in summer, "feed" and find their energy in the warm waters of the oceans. The IPCC also argues there is no certainty about storms in extra-tropical latitude. In addition, in the North Atlantic Ocean, tropical and extra-tropical cyclones seem to increase more in intensity than frequency (IPCC, 2014), mostly due to internal climate variability such as NAO than global warming. This internal climate variability could result in a greater variety of intensity and trajectories of North Atlantic storms rather than increase in frequency.

---

## Referee Comment (RC2) · H. Regnauld (Referee) · 27 Aug 2018

some sentences seem to be a dialogue between the two authors and should probably be supressed Line 161 ; 166-167, 176, 192, 216,263,269,289,

some papers have been published , dealing with palaeo climate during the LIA in the region of Laval (mainland, but close to the coast) They may be of some use for you : hal-01150657v1 Article dans une revue Jean-Pierre Marchand, Valerie Bonnardot, Olivier Planchon.. Le climat de Laval au début de la Renaissance.- Essai de géographie historique. Annales de Bretagne et des Pays de l Ouest, Presses Universitaires de Rennes, 2015, 122 (1), pp.103-133

hal-01345901v1 Chapitre d'ouvrage Jean-Pierre Marchand, Olivier Planchon., Valerie Bonnardot. La variabilité des types de temps mensuels au XVIIIème siècle à Laval : approche méthodologique. FALLOT Jean-Michel, JOLY Daniel, BERNARD Nadine. Climat et pollution de l'air., UNIL et UBFC, pp.315-320, 2016, Actes du XXIXème colloque de l'AIC, Lausanne-Besançon (6-9 juillet 2016)., 978-2-907696-22-7

But this are not hard criticisms : your paper is a very interesting one and should be published. Bravo

---

## Author Comment (AC2) · 3 Sep 2018

Dear Referee Regnauld H.,

First of all, thank you for your review on our paper Âń Vulnerability, resilience and adaptation of societies during major extreme storms during the Little Ice Age Âż.

And most importantly, thank you for your very attentive reading ! You were absolutly right for the lines of dialogue. We forgot to delete them, we will do it.

Regarding the 2 papers you mentionned, they are of great interest and useful for our reflection. Thank you for that too.

---

## Author Response (AR1)

**Author's response**

First of all, authors thank reviewers and handling editor for their reviews and advices.

Except for removed sentences, all the modifications appear in yellow in the modified manuscript.

1) As required and needed, the discussion remarks in the text have been removed.
2) The expression « climate hazards » have been replaced by « meteorological hazards » to talk about storms has recommended by reviewer 1.
3) Two references have been added as required lines 225 and 298-299.
4) Regarding the maps, the frame has been modificated, the Cotentin Peninsula is not part of the study.
5) Few reflexions on figures 2, 3 and 4 have been added in order to discuss them in greater depth.
6) A slight extension to the abstract has been provided, as same as the conclusion.
7) We informe that we have to remove the figure 5 (Map of the Champagné swamps) since we could not obtain in time the autorisation of publication from the Archives.
8) Problem of lack of clarity on lines 47 to 53 has been treated. The point was not on a parallele between climatic periods and storms but on societies adaptation and how past societies can be of inspiration for present ones. As it was not clear and we already expose it later, we removed part of the sentence.

[revised manuscript text omitted]

---

## Author Response (AR2)

**Author's response 2**

First of all, authors thank you for pointing needs in technical corrections. They appear in yellow on the modified manuscript :

1) Informations and specifications about LIA (from line 46 to 52) have been added.

2) The occurrences Little Ice Age have been replaced by LIA (lines 68, 112, 154).

3) Few lines about results regarding the storm impact on population and reduction of vulnerability have been added in the conclusion.

4) On fig. 2, the note on the bar of the 1460s has been removed. We also modificated the name « First HYPER-PAG », PAG being the french acronym, by « First HYPER-LIA » and we realised the existence of a typo on « Beautiful XVIth » (it was written Beautiful XIVth, which is not the same !).

5) On fig. 3, we changed the orthographe of Saint-Maixant, which was wrong to « Saint-Maixent ». Moreover, behind « Saint-Maixent » it was written in parentheses l'Ecole. Saint-Maixent-l'Ecole is the actual name of the ancient city Saint-Maixent. That is the reason why it appeared in parentheses. But since it is not useful for the understanding, we removed it.